# Improved maximum growth rate prediction from microbial genomes by integrating phylogenetic information

Liang Xu [1] ✉, Emily Zakem[1] & JL Weissman [2,3] ✉

Microbial maximum growth rates vary widely across species and are key parameters for ecosystem modeling. Measuring these rates is challenging, but genomic features like codon usage statistics provide useful signals for predicting growth rates for as-yet uncultivated organisms. Here we present Phydon, a framework for genome-based maximum growth rate prediction that combines codon statistics and phylogenetic information to enhance the precision of maximum growth rate estimates, especially when a close relative with a known growth rate is available. We use Phydon to construct a large and taxonomically broad database of temperature-corrected growth rate estimates for 111,349 microbial species. The results reveal a bimodal distribution of maximum growth rates, resolving distinct groups of fast and slow growers. Our work provides insight into the predictive power of taxonomic information versus mechanistic, gene-based inference.

Microbes are crucial players in global nutrient cycles, and their maximum growth rates are key parameters in ecosystem models[1–3]. Traditionally, maximum growth rates are inferred from measurements using laboratory isolates or field-based methods, such as nutrient uptake experiments[4,5] and peak-to-trough measurements[6,7]. However, accurately measuring rates for many species poses significant challenges in both laboratory and field settings[8]. Only a small fraction (less than 1%) of bacterial and archaeal species from any given environment has been successfully cultured[9,10]. Even among these cultured species, maximum growth rates vary widely, with population doubling times ranging from minutes to days across species and culture conditions[9,11–13], adding further complexity to measurement and cultivation efforts.

As a powerful alternative to cultivation for hard-to-grow species, genomic information can be leveraged to estimate the maximum growth rate of an organism. Maximum growth rate—how fast a population can grow under optimal conditions—is a broad indicator of an organism's overall evolutionary strategy and a key parameter for describing population dynamics. Several genomic features have been linked to maximum growth rates, including rRNA operon

copy number[14–17], tRNA multiplicity[18,19], replication-associated gene dosage[20,21] and codon usage biases (CUB)[18,22,23]. Among these, CUB has shown the strongest correlation with growth rates[11,12]. In fast-growing species, highly expressed genes tend to preferentially use certain synonymous codons, a bias that arises from the need for efficient translation. This optimization ensures the rapid production of proteins[22]. The robustness of CUB has been demonstrated even when extrapolating predictions across different phyla[11].

Although codon usage bias (CUB) is a widely used genomic predictor of maximum growth rates, the resulting estimates can still exhibit considerable variance and bias[11]. This inaccuracy may stem in part from the fact that traits such as growth are influenced by multiple genetic factors while CUB captures only one aspect of this complexity. Therefore, while CUB reflects evolutionary optimization for rapid translation and, by extension, rapid growth, its precision in estimating growth rates is theoretically limited. To improve accuracy, additional signals can be incorporated. One such signal is phylogenetic relatedness: closely related species tend to exhibit similar trait values due to their shared evolutionary history[24] and vertical gene inheritance[25]. This phenomenon, known as phylogenetic signal[26,27],

[1]Department of Global Ecology, Carnegie Institution for Science, Stanford, CA, USA. [2]Institute for Advanced Computational Science, Stony Brook University, Stony Brook, NY, USA. [3]Department of Ecology and Evolution, Stony Brook University, Stony Brook, NY, USA. ✉e-mail: lxu@carnegiescience.edu; Jackie.weissman@stonybrook.edu

offers a complementary approach to CUB and can help improve accuracy in CUB-based growth rate predictions.

The simplest model for trait inference from phylogenetic trees estimates a species' trait based on the trait of its nearest neighbor in the phylogenetic tree, leveraging the tendency for phylogenetically related organisms to exhibit similar phenotypes[28]. More advanced approaches involve specifying models of trait evolution, commonly using a Brownian motion framework, where trait values can be estimated for a query species based on its position in the tree and distance to neighboring species[24,28–35]. More sophisticated trait evolution models, beyond the Brownian motion model, exist[32–34,36], but they often require additional information, such as convergent trait values or eco-evolutionary timescales. Of course, the accuracy of all phylogenetic prediction methods generally decreases as phylogenetic distance increases, depending on how strongly conserved the trait is across evolutionary time.

Microbial traits vary widely in their degree of phylogenetic conservation. Martiny et al.[27] developed a phylogenetic metric to quantify this conservatism and estimate the clade depth at which organisms share a given trait. They found that over 90% of functional traits in their data were significantly non-randomly distributed. However, the clades sharing these traits were generally shallow, suggesting a moderate degree of phylogenetic conservatism. As a result, the accuracy of utilizing phylogenetic structure to estimate trait values remains uncertain and may vary by trait and context[37]. Typically, complex traits involving multiple genes (e.g., photosynthesis or methanogenesis) tend to exhibit stronger phylogenetic conservatism than simpler traits, such as the consumption of a specific carbon source[27,37]. This raises the question of whether phylogenetic relationship can reliably predict maximum growth rates, which are determined not by any one gene or set of genes, but rather as a complex outcome of a variety of genomic factors. Walkup et al.[26] assessed phylogenetic-based predictions of bacterial growth rates across environments and found that phylogenetic relationships accounted for only 38% variation in maximum growth rates across ecosystems. Moreover, such tools will only work well when a high-quality reference database of species with known trait annotations already exists for a given environment. Thus, the effectiveness of phylogenetic prediction methods depends heavily on the quality of trait data and phylogeny, as well as the strength of the phylogenetic signal.

In this study, we aim to enhance the accuracy of estimating maximum growth rates by integrating codon usage bias (CUB) with phylogenetic relatedness to create a hybrid approach for trait prediction. We evaluated the performance of a genomic CUB-based method (gRodon[11];) against two phylogenetic prediction models: the nearest-neighbor model (NNM) and the phylogenetic independent contrast-based Brownian motion model (Phylopred). To ensure robust evaluation, we tested model performance across a range of phylogenetic distances via cross validation. To make use of information from both CUB and phylogenetic relationship among species, we developed a novel R package, Phydon, which synergistically combines both approaches. Our results demonstrate that Phydon improves the accuracy of maximum growth rate estimations for microbial genomes, particularly for faster-growing organisms and when a close relative with a measured maximum growth rate is available.

## Results and discussion
### A phylogenetically informed model for maximum growth rate prediction

We compiled a dataset of 633 species with recorded doubling times from the Madin et al. trait database[9]. However, 85 species were excluded due to unidentifiable species names in the Genome Taxonomy Database (GTDB). As a result, our final dataset comprised 548 species (Fig. S2). The maximum growth rates of the species in our dataset exhibit a moderate phylogenetic signal, as indicated by a Blomberg's $K$ statistic[28] of 0.137 and a Pagel's $\lambda$ statistic[38] of 0.106 with $p$-value < 0.0072 for bacteria species, and a Blomberg's $K$ statistic of 0.0817 and a Pagel's $\lambda$ statistic of 0.17 with $p$-value < 0.0055 for archaea species. For reference, a value approaching or exceeding 1 indicate strong phylogenetic conservatism, while a value of 0 indicates no phylogenetic signal. These values suggest that while there is some degree of phylogenetic conservatism (Fig. 1), it is not overly strong. This makes the dataset well-suited for developing a method that balances genomic and phylogenetic factors. We further explored how different prediction methods perform under varying conditions to identify the most effective approach for improving growth rate predictions.

The phylogenetic distance between the training and test datasets is a critical factor in evaluating the performance of the two methods. To assess this, we successively divided the phylogenetic tree into two groups (training and test) based on varying phylogenetic distances, which is a variant of the phylogenetic blocked cross-validation analysis[39] (Fig. 2). A cutting time point $D_c$, at which the tree is divided into several clades, is identified based on the desired number of

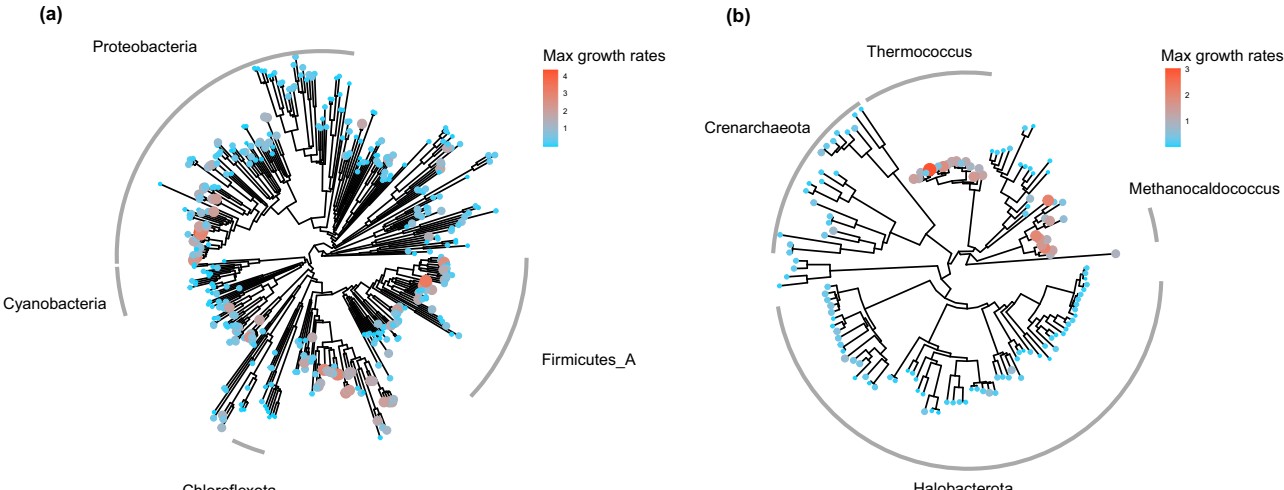

**(a)** Proteobacteria · Max growth rates · Cyanobacteria · Firmicutes_A · Chloroflexota

**(b)** Thermococcus · Crenarchaeota · Max growth rates · Methanocaldococcus · Halobacterota

**Fig. 1 | Phylogenetic conservation of maximum growth rate.** Phylogenetically conserved patterns in the maximum growth rates of bacteria (**a**) and archaea (**b**). Among bacteria, certain groups within *Proteobacteria* and *Firmicutes_A* exhibit

higher growth rates, while in archaea, the genera *Thermococcus* and *Methanocaldococcus* are notable for their rapid growth.

**(a)**

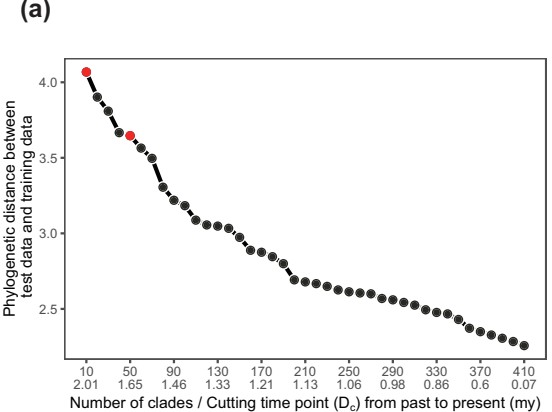

**(b)**

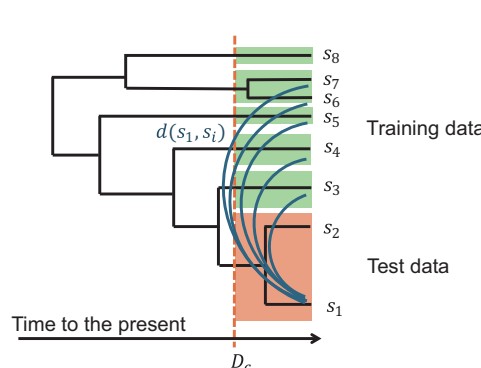

- $D_c$ is the cutting time point at which the tree is separated into $n$ clades (here $n = 6$).
- $D_p = \frac{1}{5}\sum_{i=3}^{7} d(s_1, s_i)$ is the average phylogenetic distance between a species ($s_1$) in the test data and 5 nearest species in the training data.

**(c)**

**(d)**

**Fig. 2 | Model training and cross validation. a** The phylogenetic distance between the training and test datasets is defined as the minimum average phylogenetic distance ($D_p$) between species in the test set and those in the training set. This distance decreases as the number of clades increases when the tree is cut at time points $D_c$ closer to the present. **b** $D_c$ represents the cutting time point at which the phylogenetic tree is divided into $n$ clades. For cross-validation, we iteratively use each clade as the test dataset while treating the remaining clades as the training dataset. **c**, **d** Phylogenetic trees cut at two different time points, resulting in 10 and 50 clades, respectively, are illustrated to demonstrate blocked cross-validation.

clades, $n$ (Fig. 2). Cutting the tree closer to the present results in a greater number of clades with smaller phylogenetic distances between them, while cutting further in the past at larger time points produces fewer clades with greater phylogenetic distances (Fig. 2a). This cutting time point thus serves as a proxy for the phylogenetic distance between training and test clades. This is a form of phylogenetically blocked cross-validation, wherein observations are grouped into folds following their evolutionary relationships rather than at random[39]. We trained models on each training dataset, and their performances were evaluated using each test dataset respectively. For instance, cutting the tree at the time point of 2.01 $my$ results in 10 clades. We iteratively designated one clade as the test data while using the remaining clades as the training data, thereby training a total of 10 models. The performance of each model was evaluated on its corresponding test data, and the mean squared error (MSE) scores were averaged to determine the overall MSE for this cut (Fig. 2). In doing so, the ability of each model to extrapolate to new taxonomic groups not in the training data was tested directly and thoroughly. For further details on the analysis design, we refer readers to the Methods section.

The gRodon model generally distinguishes fast and slow-growing species. Its performance is consistent across the tree of life, as demonstrated by a stable mean squared error across varying phylogenetic distances (Fig. 3a). This finding supports the notion that codon usage bias serves as an effective genomic proxy for bacterial growth rates[11,12], capturing selective pressures on genomes over evolutionary time[18,19,22,40]. Additionally, our cross-validation analysis (see Methods, Fig. 2) indicates that the relationship between CUB and the maximum growth rates generalizes well across different clades. However, we also

observed significant variance in the growth rate estimates, which persists even with decreasing phylogenetic distance between training and test sets (Fig. S1). This suggests that while CUB is a valuable predictor, additional factors beyond codon bias influence bacterial growth rates.

Phylogenetic prediction methods show increased accuracy as the minimum phylogenetic distance between the training and test sets decreases. As shown in Fig. 3a, the mean squared error (MSE) for both the NNM and Phylopred models decreases significantly as the minimum phylogenetic distance between the training and testing data narrows from the cutting time point 2.01 $my$ to 0.07 $my$. We identified cutting time thresholds below which the MSE of these phylogenetic models falls below that of the gRodon model. We also observed that Phylopred and NNM have distinct thresholds, with Phylopred showing more stable and superior performance. Based on this, we chose the Phylopred model to develop a combined approach with the gRodon model.

Interestingly, we observed divergent performance patterns between the gRodon and Phylopred models for fast-growing and slow-growing species (Fig. 3b, c). For slow-growing species, the gRodon model consistently outperforms the Phylopred model across all phylogenetic distances (Fig. 3c). In contrast, the Phylopred model shows superior performance over the gRodon model for fast-growing species as the phylogenetic distance decreases (Fig. 3b). At the smallest cutting time ($D_c = 0.07my$), Phydon reduced the median squared error for species with doubling times under 30 min by 22.4% compared to gRodon (Fig. 3a, inset). These findings suggest that the cutting time threshold or the phylogenetic distance between test and training data

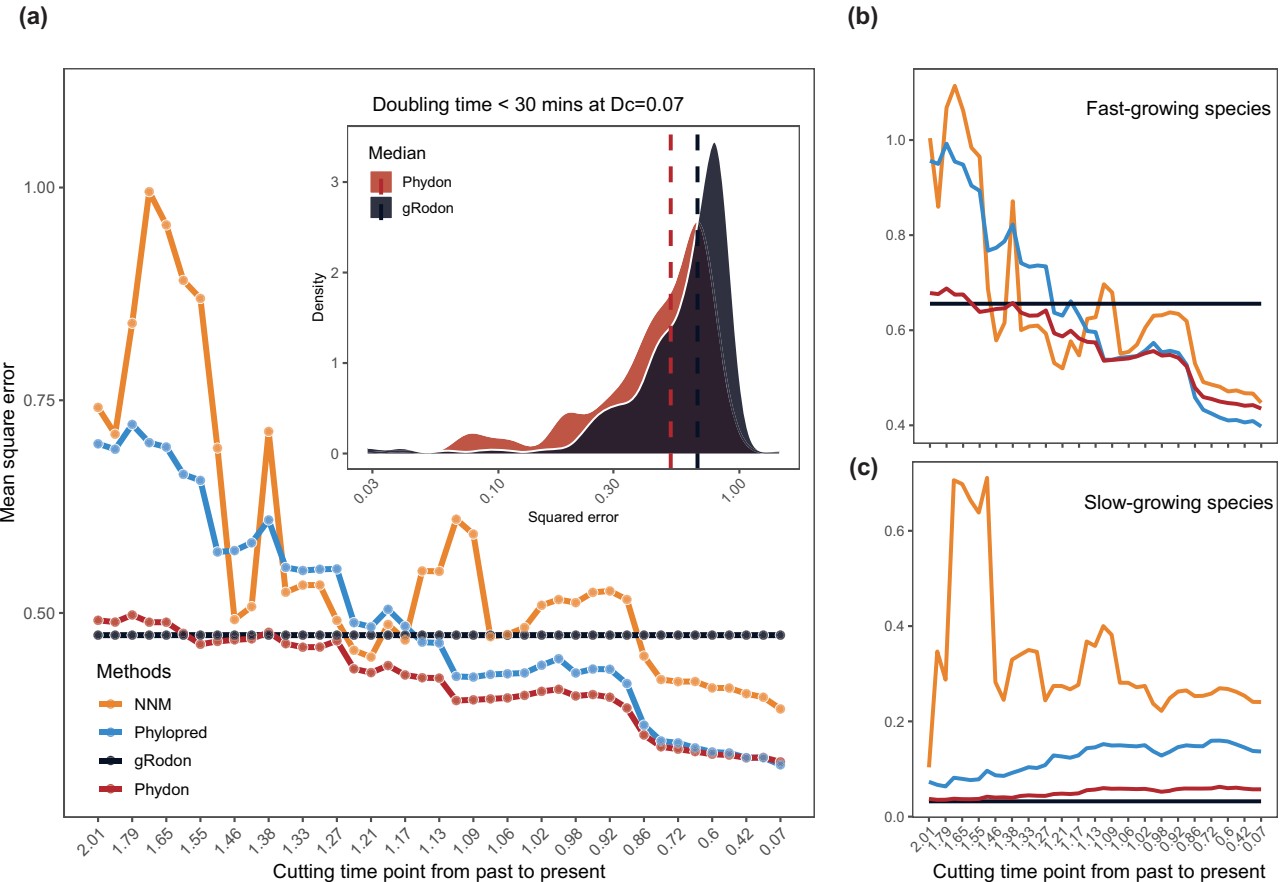

**Fig. 3 | Comparison of the mean square error of the four methods. a** The prediction error of the four methods varies with phylogenetic distance. The inset figure shows the distribution of squared error of Phydon and gRodon models at the smallest phylogenetic distance for species of doubling time <30 minutes; (**a**) Blocked cross-validated error estimates for each method when folds for cross-validation were determined phylogenetically with the cutoff of phylogenetic distance varied; **b, c**) the MSE scores for the slow-growing species (the doubling time > 5 h) and fast-growing species (the doubling time < 5 h).

sets may differ for fast- and slow-growing species. Consequently, both growth rates and phylogenetic relationships should be considered, rather than relying on a single threshold. Additionally, the results suggest that, for fast-growing species, phylogenetic relationships capture selective signals more effectively than codon usage bias, which is inherently limited as a genomic statistic for explaining maximum growth rates[41]. However, both methods face challenges in predicting traits for slow-growing species. One challenge is the difficulty of culturing these species and accurately measuring their maximum growth rates, leading to insufficient and potentially low-quality data for model training.

The lower mean squared error (MSE) achieved by the gRodon model compared to the Phylopred model may be due to the distribution of slow-growing species across the phylogenetic tree. These species exhibit weak phylogenetic signal, which challenges Phylopred's reliance on evolutionary relationships. In contrast, the CUB patterns from slow-growing species in the training dataset remain informative for predicting traits in the test dataset, enhancing gRodon's performance.

Based on the analysis of CUB- and phylogenetic distance-based models, we sought to build a predictive model that combined the strengths of these two approaches, taking into account which model would most likely work best for a given organism. To achieve this, we developed a weighted predictor where the weight of each model was determined by both the distance of the query genome to the model training set and by the gRodon estimate of growth rate (as a rough estimate of whether an organism was likely to be a fast or slow grower, see Methods). Thus, our weighting scheme takes into account how the relative expected accuracies of gRodon prediction and phylogenetic prediction change with both the phylogenetic distance of the query genome to the training set and the expected growth rate of the query genome (Fig. S6). This ensemble model, implemented in the R package Phydon, demonstrates superior predictive accuracy compared to the individual models on average (Fig.3). Specifically, the Phydon model achieves lower Mean Squared Error (MSE) scores under most of the phylogenetic distances, while similar MSE scores compared with that of the gRodon model were observed when phylogenetic distances are large. Specifically, Phydon achieves a lower MSE -- a 31% reduction in MSE -- as compared to mean square error of gRodon. We also note that the difference in performance (MSE) between Phydon and gRodon at short distances is 8.54 times the difference in performance at long distances, indicating a significant improvement at short distances with little compensation at long distances. Additionally, the variance of the Phydon's predictions is lower than that of alternative phylogenetically-aware prediction approaches, although the variance of the gRodon model is nearly indistinguishable from that of the Phydon model (Fig. S1).

We defined the weight parameter $P$ as a continuous value between 0 and 1 (see Method), ensuring that Phydon estimates always incorporate information from both phylogenetic relationships and genomic statistics. The parameter $P$ can also be treated as a binary variable, selecting the method that achieves higher accuracy at a given phylogenetic distance. However, similar to known statistical challenges with piecewise regression this approach introduces instability in overall performance due to uncertainty when estimating the appropriate threshold value for switching between models (Fig. S5). Continuous weighting schemes (as used above) average over such uncertainty. Our results demonstrate that arithmetic models with a continuous $P$ outperform those with a binary $P$ (Fig. S5). Alternatively, the binary $P$ approach leads the model to discard information from one source entirely, favoring either phylogenetic relationships or genomic statistics, but never both. Thus, we disfavored such an approach.

## A comprehensive growth rate database for amplicon analysis

While multi-omic methods for surveying microbial communities in the environment have rapidly matured, amplicon sequencing, typically of the 16S rRNA gene, remains a cost-effective and widely used approach for assessing microbial diversity[42]. Various tools exist to link functional annotations to amplicon sequencing data, though the quality of these annotations varies widely depending on the taxonomic group and trait of interest[43–45]. Annotation quality depends on (1) the degree of trait conservation between closely related organisms, and (2) the comprehensiveness of the associated trait database used for functional annotations. Given its moderate phylogenetic conservation (see above), maximum growth rate is a suitable candidate for database-assisted functional annotation[11]. Yet, database quality remains a limiting factor.

Previously, the EGGO database of gRodon annotations addressed some of the challenges associated with functional annotation[11]. Yet, EGGO was primarily comprised of genome annotations from lab-cultivated organisms and lacked optimal growth temperature corrections, which are crucial for accurate gRodon predictions. To address these limitations, we developed an improved maximum growth rate database by 1) annotating species representative genomes from GTDB v220, which includes a majority of metagenome-assembled genomes (MAGs) and single-cell amplified genomes (SAGs)[46], 2) incorporating temperature corrections using genomic optimal growth temperature estimation software[47], and 3) applying our Phydon predictor for improved estimation. The rigorously curated GTDB provides the additional benefit of high-quality taxonomic annotations, which enhances the reliability of downstream analyses.

We ran Phydon on all 113,104 GTDB v220 species representative genomes, with 111,349 passing quality filters needed for internal gRodon prediction (e.g., having at least 10 annotated ribosomal proteins[11]). Of these, we were able to annotate 111,034 genomes with optimal growth temperature predictions from GenomeSPOT[47] and subsequently pass them through Phydon. This database includes 111,034 temperature-corrected maximum growth rate predictions. Of the 111,034 species in this database, a total of 60,869 had genomes with 16S rRNA genes present (16S rRNA recovery often fails for MAGs[48–50]), representing 17,451 genera and 191 phyla (Fig. 4). To facilitate access for researchers to this database, we provide an online tutorial alongside the Phydon package (https://github.com/xl0418/Phydon).

Phydon detects major divisions in growth strategy between microbial phyla (Fig. 4a), consistent with our understanding of the typical lifestyles and metabolisms associated with these groups. For example, the mostly-heterotrophic *Bacillota* (*Firmicutes*) tend to be fast-growers whereas oxygenic phototrophic *Cyanobacteria* and sulfate-reducing *Desulfobacterota* bacteria tend to be slow-growers. Our findings also reveal a clear bimodal distribution in the estimates of maximum growth rates for species in the GTDB (Fig. 4b), consistent with previous observations using the gRodon package[11]. Interestingly, Phydon and Phylopred both seem to distinguish these major growth classes much more readily than gRodon (Fig. 4b). As we would expect, the predictions of Phydon and gRodon converge as phylogenetic distances increase (Fig. 4c), showing how Phydon increasingly relies on genomic factors, particularly codon usage bias (CUB), for extrapolation.

Genomic and phylogeny-based methods each have distinct advantages for trait prediction under different scenarios. Specifically, the CUB-based gRodon model excels at predicting the maximum growth rates of phylogenetically distant species. In contrast, phylogenetic prediction models outperform the gRodon model for phylogenetically related species. Thus, incorporating phylogenetic information enhances maximum growth rate estimation more effectively than relying solely on genomic signatures like CUB. By integrating phylogenetic context, the Phydon model achieves lower error, and more specifically lower variance than the other methods when used individually.

However, our method is not without limitations. A key challenge lies in its performance when applied to taxa that have undergone rapid trait evolution. Periods of accelerated trait evolution, such as those occurring in rapidly changing environments, can weaken the predictive

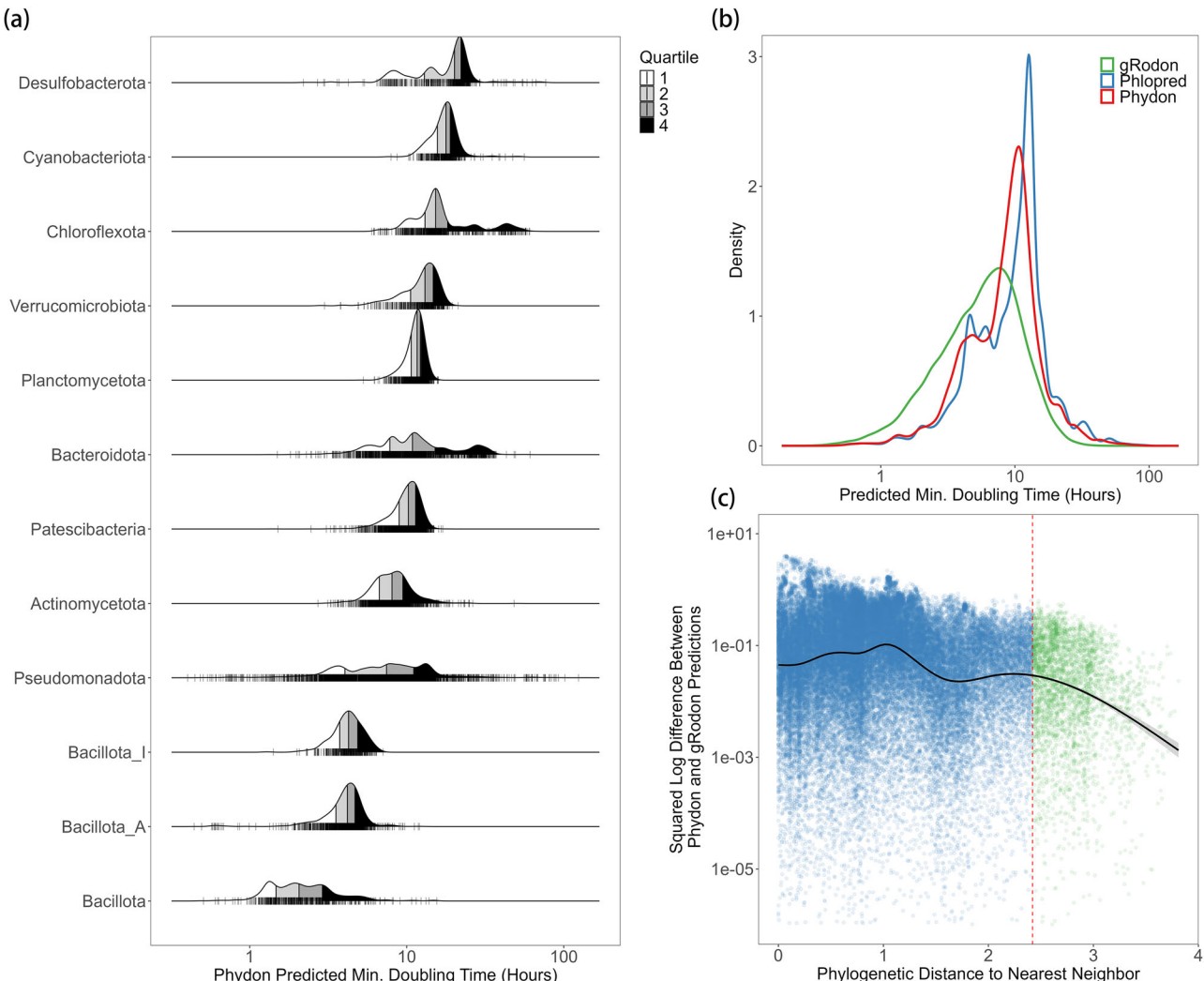

**Fig. 4 | The distribution of estimated maximum growth rates of major phyla from GTDB. a** The distribution of the Phydon predicted maximum growth rates of major phyla (at least 1000 species representative genomes in GTDB v220; temperature corrected using genomic optimal growth temperature estimates). **b** The distribution of the estimates of maximum growth rates from Phydon and gRodon and Phylopred. **c** Phydon and gRodon predictions converge for distantly related organisms. Dashed vertical line at a phylogenetic distance of 2.42 corresponding to where gRodon and Phylopred have approximately equal performance.

power of both phylogenetic models and models based on genome-scale evolutionary patterns like CUB. These models rely on the assumption of relatively stable evolutionary processes[51], and this represents an ultimate limitation of all genome-based trait prediction models.

Despite this, our predictions offer a useful indication of an organism's ecological niche, reflecting a long-term integration of evolutionary trends. It is important to interpret predicted maximum growth rates with this context in mind, as instantaneous growth rates can vary significantly over time and space depending on the local environmental conditions and the organism's state.

A principled approach to model design involves recognizing the limitations and potential failure points of each model. By understanding where models are likely to encounter difficulties, researchers can strategically design complementary studies and integrate diverse methodologies to offset these weaknesses. Our integrated approach to trait prediction leverages phylogenetic prediction for closely related organisms and genomic model-based prediction to extrapolate to groups with limited representation in the reference database. This strategy provides a pathway for high-quality genome annotation with trait information, offering a robust solution for trait prediction for uncultured microbes. For example, new tools have recently been developed to estimate growth temperature range, aerobicity, optimal

pH, optimal salinity, and other traits from diverse genomic signals, including amino acid usage patterns[47,52]. Each of these genomic trait predictors can potentially be embedded in a Phydon-like model that averages their output with a straightforward phylogenetic prediction to obtain a more accurate prediction.

In sum, Phydon provides accurate maximum growth rate predictions for uncultured taxa, enabling scientific exploration of the ecological roles of uncultured microbial majority and their principled incorporation into ecosystem models. Phydon serves a dual role as both a powerful approach for hypothesis generation for microbial scientists working across systems (e.g., seawater, soils, and the human gut) and a key source of information for modelers who are increasingly interested in developing whole-community models of microbiome dynamics in environmental and host-associated systems[53–55]. More broadly, our work provides insight into the line at which taxonomic information surpasses mechanistic, gene-based inference, speaking to the balance between empirical vs. theoretical and the ability of each to yield predictive power in microbiology. This threshold, which varies across the tree due to local changes in evolutionary rate and environmental heterogeneity, can be used as a metric to investigate differences in predictability among bacterial guilds and across the prokaryotic taxonomic landscape.

## Methods

### The training datasets and phylogeny

For the 548 bacteria and archaea species in our data set, we fetched bacterial and archaeal phylogenetic trees from GTDB r220[46,56,57]. Figure 1a presents the final phylogenetic tree for bacteria (411 species) alongside the corresponding trait distribution, while the phylogenetic tree for archaea (137 species) is presented in Fig. 1b. To assess phylogenetic conservation, we calculated the pairwise differences in maximum growth rates for all species and plotted these against their phylogenetic distances. Figure S3 demonstrates that species with close phylogenetic relationships exhibit smaller trait differences, whereas more distantly related species show larger differences, highlighting the phylogenetic conservation of maximum growth rates.

Temperature is a critical factor in regulating microbial growth, and incorporating optimal growth temperature into codon usage bias (CUB)-based models can significantly enhance the accuracy of maximum growth rate predictions. We gathered data on the optimal growth temperatures of species from the Madin et al. database[9]. Of the 411 bacterial species in our dataset, 374 had recorded optimal growth temperatures, while 69 out of 137 archaeal species also had this data available. The gRodon R package (version 2.4.0) features two models: one trained with temperature data and one without. To evaluate the impact of incorporating temperature on growth rate predictions, we conducted analyses using both models and compared their predictive performance to assess the enhancement provided by including optimal temperature information.

Each species in GTDB is associated with multiple genomes, with one designated as the representative genome and the remainder clustered within the GTDB phylogenetic tree. To balance information content and computational efficiency, we performed random sampling, selecting up to five genomes per species in our training set (or all available genomes if fewer than five were present) while always including the representative genome. We assume that highly similar genomes share traits. This assumption is essential to our methods and also to other prediction methods that use genomic statistics, where it has been shown that maximum growth rate is strongly conserved up to approximately the genus level[11,12]. This approach resulted in a dataset comprising 1465 complete genomes for our 411 bacteria species and 323 genomes for 137 archaea species.

### Assessing the effect of phylogenetic relatedness on the gRodon model and phylogenetic models

Two phylogenetic prediction models were used as benchmarks to determine the conditions under which the gRodon model outperforms traditional phylogenetic prediction methods. The first phylogenetic prediction model, known as the nearest neighbor method (NNM), estimates the maximum growth rates of focal species in the testing data by averaging the maximum growth rates of the most phylogenetically related sister species in the training data. To identify the optimal number of related species for NNM, we tested groups of 1, 5, 10, 20 and 50 species. Our analysis revealed that averaging the traits of the 5 closest phylogenetic species resulted in the lowest mean squared error (MSE), although differences among group sizes were minimal (Fig. S4). Thus, we used the average trait of the 5 closest species in the NNM method.

The second phylogenetic prediction model, referred to as the phylopred method, predicts maximum growth rates of species using Bloomberg's *K* statistics and Felsenstein's independence contrast (IC) via the *phyEstimate* function from the picante package (version 1.8.2)[28,31]. This approach calculates the mean maximum growth rates of the most phylogenetically related species, with each species trait weighted according to its phylogenetic distance from the query genome. Essentially, phylopred is a weighted variant of the nearest neighbor method, where the contribution of each neighbor is adjusted based on its phylogenetic distance to the query species.

We used phylogenetically blocked cross-validation to evaluate the performance of both the gRodon model and phylogenetic prediction models[39]. We divided the phylogeny into *n* clades by cutting the tree at a uniform depth (Fig. 2). As *n* increases, the average phylogenetic distance between each pair of clades decreases (*i.e.*, the depth at which we cut in the tree becomes shallower), meaning that for each test set the nearest clade in the training set becomes more closely related on average (Fig. 2a). We re-trained the gRodon model using the same approach as for Phydon, i.e., training models on genomes from *n* − 1 clades and tested it on genomes from the remaining clade (Figure 2b, c). This process was repeated across *n* values ranging from 10 to 410 in increments of 10. When *n* = 10, there is a large phylogenetic distance between the training and test clades, while *n* = 410 represents a small minimum phylogenetic distance to the nearest clade (Fig. 2). This design created 41 test scenarios to assess model performance under varying degrees of phylogenetic relatedness.

### Phydon: Combining the gRodon model and phylogenetic prediction models

To leverage the complementary strengths of the gRodon and phylogenetic prediction models for forecasting maximum growth rates, we developed a novel combined regression model, named Phydon[58]. This model integrates predictions from both approaches, calculating the maximum growth rate as a weighted mean of the gRodon and phylogenetic predictions. Phydon operates in two modes:

Arithmetic Mean Mode: This mode calculates the Phydon prediction as:

$$\widetilde{y}_{\text{phydon}} = \widetilde{y}_{\text{gRodon}} \times P + \widetilde{y}_{\text{phylopred}} \times (1 - P) \tag{1}$$

where $P$ represents the probability that the gRodon model outperforms the phylogenetic models.

Geometric Mean Mode: This mode uses the geometric mean of predictions from the two models:

$$\widetilde{y}_{\text{phydon}} = \widetilde{y}_{\text{gRodon}}^{P} \times \widetilde{y}_{\text{phylopred}}^{1-P} \tag{2}$$

We suspected that the geometric mean model would be more suitable for averaging growth predictions given its usual application for analyzing percentage changes and positively skewed data where it provides greater accuracy. Nevertheless, our results show that both weighted prediction modes performed similarly with the arithmetic model slightly better in its MSE score (Fig. S5). So, we presented the main results using the arithmetic mean model and set this as the default mode in the Phydon R package. The probability $P$ is determined using a regression model that considers the growth rates of the species and the average phylogenetic distance ($D_p$) between the focal species and its 5 nearest relatives in the training dataset. Intuitively, whether the gRodon or phylopred model should be preferred will be dependent on both how distant the query genome is from the model training data, as well as if that organism is a fast or slow grower (since model performance varies with growth rate for both approaches). The regression model, based on logistic regression using the glm function in R, is given by:

$$\text{logit}(P) \sim \log(\widetilde{y}_{\text{gRodon}}) + D_p + \log(\widetilde{y}_{\text{gRodon}}) \times D_p + \varepsilon \tag{3}$$

Here, logit($P$) represents the log-odds of the gRodon model being superior, $\widetilde{y}_{\text{gRodon}}$ is the gRodon prediction, and $D_p$ is the average phylogenetic distance between the query genome and the 5 nearest species in the training data set. Given that the true growth rate of the target species is unknown, the gRodon prediction serves as a proxy for growth rate in Eq. 3, and thus allows our weighting scheme to account for how the expected relative performances of gRodon and Phylopred change with the maximum growth rate of the query organism. We then

compared the performance of Phydon with both the gRodon model and the phylogenetic prediction models to evaluate its efficacy.

We also assessed the performance of the models with a binary $P$, given by

Mode with a binary P: This mode calculates the Phydon prediction as:

$$\widetilde{y}_{phydon} = \begin{cases} \widetilde{y}_{gRodon} & P > 0.5 \\ \widetilde{y}_{phylopred} & P \leq 0.5 \end{cases} \quad (4)$$

where $P$ represents the probability that the gRodon model outperforms the phylogenetic models.

### Phydon prediction database

We annotated all GTDB v220 species representative genomes[46] using prokka[59] and GenomeSPOT[47]. We then passed these genomes and their optimal temperature predictions from GenomeSPOT to Phydon. For genomes with $\Psi > 0.6$ (see ref. [60]), indicating possible contamination, we ran Phydon using gRodon's metagenome mode.

### Reporting summary

Further information on research design is available in the Nature Portfolio Reporting Summary linked to this article.

## Data availability

The maximum growth rate data of species used in this study are publicly available from Madin et al[9]. The phylogenetic data and genomic data used in this study are available in GTDB v220[46]. No new data was generated in this study, but growth rate predictions for the EGPO genomes are available in the associated github repository alongside all the data used in model training and testing[58] (https://doi.org/10.5281/zenodo.15115834).

## Code availability

The Phydon package is available at https://github.com/xl0418/Phydon. The analysis code is archived at https://github.com/xl0418/PhydonAnalysis. The code is also available at[58] (https://doi.org/10.5281/zenodo.15115834).

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

## Acknowledgements

The computations presented here were conducted in the Resnick High Performance Computing Center, a facility supported by Resnick Sustainability Institute at the California Institute of Technology. The authors would like to thank Stony Brook Research Computing and Cyberinfrastructure and the Institute for Advanced Computational Science at Stony Brook University for access to the high-performance SeaWulf computing system, which was made possible by $1.85M in grants from the National Science Foundation (awards 1531492 and 2215987) and matching funds from the the Empire State Development's Division of Science, Technology and Innovation (NYSTAR) program (contract C210148). EJZ and LX acknowledge the Carnegie Institution for Science for funding and support.

## Author contributions

L.X. and J.W. conceived the study. L.X., E.Z., and J.W. collaboratively refined the idea and designed the methods. L.X. and J.W. collected the data, conducted the experiments, and performed the data analysis. All authors (L.X., E.Z., and J.W.) discussed and refined the methods and results. L.X. developed the R package and L.X. and J.W. drafted the manuscript, which was subsequently revised and improved by E.Z.

## Competing interests

The authors declare no competing interests.
