## [Transparent Peer Review file · Nature Communications]

Improved maximum growth rate prediction from microbial genomes by integrating phylogenetic information

Corresponding Author: Dr JL Weissman

Version 0:

Reviewer comments:

Reviewer #1

(Remarks to the Author)

Xu et al. provide a novel method based on a combination of genomic and phylogenetic comparative methods to improve the prediction of the maximum growth rate across prokaryotes. The authors first measure the phylogenetic signal of the trait across 548 microbial species from a diverse and publicly available dataset. Then, they infer phylogenetic trees following a standard pipeline and incorporate the phylogenetic structure and non-independence between species into a predictive model that yields a wide range of training and test sets in terms of diversity and relatedness. Moreover, the authors benchmark different predictive models while discussing the strengths and weaknesses of each one. Finally, they applied the model to a larger dataset that contains uncultured microbes. This effort results in a database for growth rate and a ready-to-use R package, which will be invaluable to the scientific community.

We congratulate the authors on the clear, logical flow of their ideas and the quality of their writing. We endorse the publication of this manuscript, provided the authors address a few minor technical details and clarifications.

Line 282: The total number of species is 548, and the supplementary figure should be S2.

Line 354: Briefly discuss what is the advantage or disadvantage of the two modes of Phydon. In what scenarios can a user choose one over the other?

Line 369: In equation 2, please state whether D is the average distance between the query genome and the (top 5?) closest species in the training set.

Have the authors thought about using the predictions from gRdon to adjust the P parameter, which will later be used again to assign a weight to each model (gRdon vs Phylophred)?

Line 571: Although the pattern is noisy, the blue line shows that overall trend. Please indicate in the figure legend what is the blue line. Is it a polynomial fit?

Line 540: Figure 3a shows the minimum distance as 0.14 MY, yet Figure 2a shows the minimum distance for $n = 410$ clades as close to 2.5. Can you explain this disconnection between the distances from both figures? How was the 0.14 value obtained? Also, it will be helpful to mention in Figure 3 that the phylogenetic distance is an average of the top five closest species to the query genome.

In the methods section, please mention the version of the packages used.

(Remarks on code availability)

The Phydon codebase is well-structured, with detailed documentation to inform the usage of this software tool.

Reviewer #2

(Remarks to the Author)

Summary

Maximum growth rate is an important parameter for modeling microbial species. However, it is difficult to measure. Thus, prediction of the growth rate is desirable. The authors present a tool called Phydon which combines prediction using codon usage statistics and prediction using phylogenetic information into a single growth rate prediction. They show that Phydon results in better predictions than both of these techniques individually across different phylogenetic distances and growth rates. Using Phydon the authors generate a database for calculated maximum growth rates for a variety of species.

Major comments

For some values (see Fig. 3a, values with phylogenetic distance > 2.6), Phydon is outperformed by gRodon. At other times Phydon is as good as the tools it is based on. Could the better performance be reached by simply setting a threshold phylogenetic distance and either choosing gRodon or Phylopred (at least for this dataset)?

How good does the proposed strategy generalize to the prediction of other traits?

It seems to me that the merit of this research is predominantly making a good strategy available rather than generating algorithmic/methodological innovation. What is unclear to me how much impact Phydon has.

There are some inconsistencies in the manuscript. For instance, I wasn't able to find the description of the analysis nor the results of the experiment referred to in Line 301, on whether gRodon with or without temperature is better.

Similarly, the authors mention in Line 355 that two modes of weighing results of phylopred and and gRodon are given, however it is not described which one is used for the analysis.

Minor comments

line 123: maybe the authors could describe the dataset in a few words, just using "our dataset" is a bit vague and doesn't help much

line 187: Yes, Phydon seems to have lower variance than Phylopred/NNM, but only barely so for gRodon, maybe some numbers would be helpful to drive home this point (as the difference in total between Phydon/gRodon is difficult to see on the diagram)

line 280-282: swap numbers 548 and 633 -> 548 - 85 != 633

line 283: should probably be Fig. S2, not S1

Fig. 3/4c): sometimes phylogenetic distance axis goes from high to low, sometimes the reverse -> could it always go from low to high?

Questions

Fig. 2a) Why doesn't this go to 0.14, as all the other figures? How do they get values for such low distances? How many clades are there in this case?

line 105-119: paragraph also contains results? Is it supposed to be like this?

line 195: I don't get this reasoning: the threshold estimate is unclear, because gRodon/Phylopred predictions are close to each other (and also because of the data) -> so a binary switch would be just better?

line 242/4b): If this diagram shows that Phydon is better than gRodon, doesn't it also show that Phylopred is better than Phydon?

line 243-246/4c): Why this sentence/figure? Isn't it by design intended that Phydon uses gRodon for distantly related organisms? Wasn't this designed this way? Is it just regression?

line 269-277: This paragraph seems off, particularly the first two sentences do not seem to fit.

line 335: Does phylogenetically blocked cross-validation do multiple tests on the same n ? So unlike normal cross-validation?

Due to these issues, we would suggest to reject the paper. Particularly, the small benefit of Phydon, combined with the small amount of innovation motivates our recommendation.

(Remarks on code availability)

Reviewer #3

(Remarks to the Author)

(Remarks on code availability)

Reviewer #4

(Remarks to the Author)

Xu et al. propose a new method to predict microbial growth rates using both codon usage and phylogenetic information. This approach improves upon a previous method gRodon (also developed by the last author), and is applied to a large number of genomes to generate a free to use database.

The method is very well documented online. I have not tried it, but the documentation seems very detailed.

The manuscript is clear and easy to follow. I have one major criticism to make, and provide some comments below that I think could help improve the manuscript.

My major criticism is methodological. First, the authors have performed cross-validation to evaluate the performance of the two base methods. It was unclear to me whether, for a given n (the number of clades), several predictions were made, where a different clade is held out each time. Second, no cross-validation was used to evaluate the performance of the combined approach. I believe common practice would be to leave out some data to train the two base methods, that could then be used to evaluate the performance of the combined approach. Otherwise, there is a risk of overfitting.

Minor comments:

p6 : "provides accurate directional prediction" : what does directional mean here?

l169: "we observed divergent performance patterns between the gRodon and PhyloPred models for fast-growing and slow-growing species (Fig. 3bc).": this is a surprising pattern, I would have expected the opposite: that the signal of selection would be strong for species with high growth rate, which gRodon could pick. For slow-growing species, I would expect little signal in codon usage, and therefore more errors for gRodon. Could the authors provide some insight into what may be happening?

l196: "we disfavored such an approach.": it would seem like this paragraph does not quite correspond to what the authors used, i.e., the geometric model with a logistic regression to infer the P parameter. I think some rewriting would clarify what the authors really did. In addition, a plot of the P value on the y axis, and the phylogenetic distance on the x axis, could be useful to understand the behaviour of the model.

l280: "We compiled a training dataset of 548 species with recorded doubling times from the Madin et al. trait database, after discarding 85 species due to unidentifiable species names in the Genome Taxonomy Database (GTDB), resulting in a total of 633 species [9] (Fig. S1).": this sentence is not clear and makes it look like you ended up with 633 species but started with 548 species. Also, I believe you want to refer to Fig S2, not S1.

l306: "To balance information content and computational efficiency, we performed random sampling, selecting up to five genomes per species in our training set ": If I understand correctly, this means that the authors are assuming that all strains of a species, despite having some differences in their genomes, share the same growth rates. Are there reasons to expect that this is indeed the case, or should this be mentioned as a caveat? Some discussion would be useful.

l354: "Phydon operates in two modes.": I have not seen a comparison of the two approaches; perhaps a figure in the supplementary material would be useful.

Fig 4b: Phlopred -> PhyloPred

(Remarks on code availability)

Version 1:

Reviewer comments:

Reviewer #1

(Remarks to the Author)

We thank the authors for addressing all of our comments and for further explaining how the phylogenetic tree was divided and how the parameter P affects the behavior of the model. Overall, the authors have improved the quality of the manuscript. Therefore, we endorse the current version for publication, provided the authors address two additional requests.

1. Adding Figure S6 to the GitHub repo will help users identify the optimal value of P for their analyses.

2. Can the authors state in the main text the purpose of the inset from Figure 3A? The MSE of Phydon is lower than that of gRodon at $D_c = 0.07$. Therefore, it is unclear to me why the authors added the inset. What message are they trying to convey?

(Remarks on code availability)

Sound.

Reviewer #2

(Remarks to the Author)

The authors substantially revised the manuscript and addressed our concerns adequately. We have only one further minor remark.

Minor comments

General remarks on the applicability of the approach to further traits let us wonder, what evidence the authors provide towards this claim. Now, a new sentence introduced in the manuscript in lines 115-117 reiterates this possibility, but unless the authors produced (preliminary) experimental evidence, it would be best to limit references to future work to Conclusion Section. While the translation to other traits is fitting for an outlook, mentioning this in the introduction could be misleading, the presented work does not as it is not the main part of the work.

(Remarks on code availability)

Reviewer #3

(Remarks to the Author)

(Remarks on code availability)

The code is well-documented and provides examples of usage.

Reviewer #4

(Remarks to the Author)

Xu et al. have satisfactorily addressed my comments, and I have no further major comments, and only minor elements below.

I congratulate the authors on their work.

Minor comments:

l75: "where traits values can be estimated": trait values

l116: "optimal growth temperature for growth": too many "growth"

l222: "at short distances while little": with little

l423: "with the arithmetic model is slightly better in its MSE score (Fig. S5)": remove "is"

(Remarks on code availability)

Reviewers' comments:

Reviewer #1 (Remarks to the Author):

Xu et al. provide a novel method based on a combination of genomic and phylogenetic comparative methods to improve the prediction of the maximum growth rate across prokaryotes. The authors first measure the phylogenetic signal of the trait across 548 microbial species from a diverse and publicly available dataset. Then, they infer phylogenetic trees following a standard pipeline and incorporate the phylogenetic structure and non-independence between species into a predictive model that yields a wide range of training and test sets in terms of diversity and relatedness. Moreover, the authors benchmark different predictive models while discussing the strengths and weaknesses of each one. Finally, they applied the model to a larger dataset that contains uncultured microbes. This effort results in a database for growth rate and a ready-to-use R package, which will be invaluable to the scientific community.

We congratulate the authors on the clear, logical flow of their ideas and the quality of their writing. We endorse the publication of this manuscript, provided the authors address a few minor technical details and clarifications.

Response: We thank the reviewer for the summary of our work. We agree that this tool will be of use to a wide range of biological scientists.

Line 282: The total number of species is 548, and the supplementary figure should be S2.

Response: The total number of species in Mardin et al.'s data is 633. We discarded 85 species as they are not identifiable in GTDB, which results in 548 species in our study. We have clarified this in the revision.

Line 121-124

"We compiled a dataset of 633 species with recorded doubling times from the Mardin et al. trait database [9]. However, 85 species were excluded due to unidentifiable species names in the Genome Taxonomy Database (GTDB). As a result, our final dataset comprised 548 species (Fig. S2)."

Line 354: Briefly discuss what is the advantage or disadvantage of the two modes of Phydon. In what scenarios can a user choose one over the other?

Response: We thank the reviewer for identifying a point that may confuse users. The geometric mean (essentially, taking the mean in log-space) is more reasonably used for growth rates where changes in population size occur multiplicatively. Nevertheless, when we directly compared the two modes, the arithmetic mean model performs

slightly better than the geometric model (S5 Fig), though the difference is minor. With this in mind, we offer advanced users both options in the R package. We also now offer a threshold-based model based on other reviewer comments. The arithmetic model is set as Phydon's default as the best-performing model under cross validation (S5 Fig), and we anticipate that most users will have no reason to change this setting.

Line 429 - 435.

"We suspected that the geometric mean model would be more suitable for averaging growth predictions given its usual application for analyzing percentage changes and positively skewed data where it provides greater accuracy. Nevertheless, our results show that both weighted prediction modes performed similarly with the arithmetic model is slightly better in its MSE score (Fig. S5). So, we presented the main results using the arithmetic mean model and set this as the default mode in the Phydon R package."

Line 369: In equation 2, please state whether D is the average distance between the query genome and the (top 5?) closest species in the training set.

Response: Yes, it is. We have used D_p to denote the average phylogenetic distance between the query genome and the 5 nearest species in the training data set and D_c to denote the cutting time point on the phylogenetic tree. We have added one sentence to clarify it and **Figure 2 b** as an illustration.

Line 445-446.

" D_p is the average phylogenetic distance between the query genome and the 5 nearest species in the training data set."

Have the authors thought about using the predictions from gRodon to adjust the P parameter, which will later be used again to assign a weight to each model (gRdon vs Phylophred)?

Response: We thank the reviewer for noting a point where we had not explained the model entirely clearly. In fact, we did exactly what the reviewer suggests, as shown in Eqn. 3. We first used gRodon predictions to give a rough estimation of the growth rates, which together with the phylogenetic distance is used to infer the P parameter in a linear model fit to the data (Eqn. 3). Then, the inferred P parameter is used to produce a weighted average of Phylopred and gRodon predictions, resulting in the final Phydon prediction (Eqns. 1 or 2). We have now clarified these points in the Methods section.

Line 206 – 213

"To achieve this, we developed a weighted predictor where the weight of each model was determined by both the distance of the query genome to the model training set

and by the gRodon estimate of growth rate (as a rough estimate of whether an organism was likely to be a fast or slow grower, see Methods). Thus, our weighting scheme takes into account how the relative expected accuracies of gRodon prediction and phylogenetic prediction change with both the phylogenetic distance of the query genome to the training set and the expected growth rate of the query genome.”

Line 446 - 451

“Given that the true growth rate of the target species is unknown, the gRodon prediction serves as a proxy for growth rate in Eq. 3, and thus allows our weighting scheme to account for how the expected relative performances of gRodon and Phylopred change with the maximum growth rate of the query organism. We then compared the performance of Phydon with both the gRodon model and the phylogenetic prediction models to evaluate its efficacy.”

Line 571: Although the pattern is noisy, the blue line shows that overall trend. Please indicate in the figure legend what is the blue line. Is it a polynomial fit?

Response: We thank the reviewer for catching this omission. We used the default setting of the plotting function `geom_smooth` in `ggplot2` in R. It uses the generalized additive model (GAM). We have added the description.

“Figure S3. The pairwise phylogenetic distance vs. the difference in log-transformed doubling time between species. The blue line is the regressed line using a generalized additive model (GAM) with the default setting in the R ggplot2 function geom_smooth”

Line 540: Figure 3a shows the minimum distance as 0.14 MY, yet Figure 2a shows the minimum distance for $n = 410$ clades as close to 2.5. Can you explain this disconnection between the distances from both figures? How was the 0.14 value obtained? Also, it will be helpful to mention in Figure 3 that the phylogenetic distance is an average of the top five closest species to the query genome.

Response: The distance on the x axis of Fig 3a is the cutting time point D_c at which the phylogenetic tree is divided to n clades while the other is the average phylogenetic distance D_p between the focal species and the 5 nearest species. Given a cutting time point D_c , the average phylogenetic distance D_p between each query species and the nearest 5 species in the training data can vary, but generally decrease with D_c approaching the present. We show in Fig. 2a that the minimum average phylogenetic distance between test and training data decreases when the tree is separated into more clades. We have added one figure (Fig 2b) to clarify those metrics and changed the axis labels accordingly in Figures 2, 3, S1, S4, and S5.

In the methods section, please mention the version of the packages used.

Response: We thank the reviewer for their detailed comments We have added

package version numbers.

Reviewer #1 (Remarks on code availability):

The Phydon codebase is well-structured, with detailed documentation to inform the usage of this software tool.

Reviewer #2 (Remarks to the Author):

Summary

Maximum growth rate is an important parameter for modeling microbial species. However, it is difficult to measure. Thus, prediction of the growth rate is desirable. The authors present a tool called Phydon which combines prediction using codon usage statistics and prediction using phylogenetic information into a single growth rate prediction. They show that Phydon results in better predictions than both of these techniques individually across different phylogenetic distances and growth rates. Using Phydon the authors generate a database for calculated maximum growth rates for a variety of species.

Response: We thank the reviewer for their summary and detailed comments.

Major comments

For some values (see Fig. 3a, values with phylogenetic distance > 2.6), Phydon is outperformed by gRodon. At other times Phydon is as good as the tools it is based on. Could the better performance be reached by simply setting a threshold phylogenetic distance and either choosing gRodon or Phylopred (at least for this dataset)?

Response: We thank the reviewer for their question, which we are sure many readers would also share. We indeed initially considered the model with a threshold-based weighting scheme but decided that given known statistical difficulties with changepoint analyses/piecewise regression such a model was likely to give highly uncertain threshold estimates. Thus, we chose continuous weighting schemes that would be less sensitive to this uncertainty and yield lower model variance overall. We have added a section to introduce the binary model (see below). As we expected, the model with a binary P shows elevated error due to increased model variance, often near where model performances intersect and thus threshold estimates will be the most variable (**Fig. S5**). The other weighting approaches do not experience these performance issues. We have added a figure (**Fig. S5**) comparing the performance of all models, such as the arithmetic model with a binary P and with a continuous P and the geometric model. We will note that while it is true that at long phylogenetic distances between training and test set gRodon does outperform Phydon slightly (we would argue marginally), it is not realistic to compare these cases to the cases where Phydon much more dramatically outperforms gRodon at shorter phylogenetic distances –Phydon achieves a lower MSE -- a 31% reduction in MSE -- as compared to mean square error

of gRodon. We also note that the difference in performance (MSE) between Phydon and gRodon at short distances is 8.54 times the difference in performance at long distances, indicating a significant improvement at short distances while little compensation at long distances.

We also do recognize that in some edge cases the model with a binary P has a slightly better performance than alternatives, for example, for species that are very close to the training data ($D_c < 0.42$). So, we have implemented this model as an option in our package available to advanced users.

We have added this discussion to the main text Lines 218-223:

“Specifically, Phydon achieves a lower MSE -- a 31% reduction in MSE -- as compared to mean square error of gRodon. We also note that the difference in performance (MSE) between Phydon and gRodon at short distances is 8.54 times the difference in performance at long distances, indicating a significant improvement at short distances while little compensation at long distances.”

Line 230 - 237

“The parameter P can also be treated as a binary variable, selecting the method that achieves higher accuracy at a given phylogenetic distance. However, similar to known statistical challenges with piecewise regression, this approach introduces instability in overall performance due to uncertainty when estimating the appropriate threshold value for switching between models (Fig. S5). Continuous weighting schemes (as used above) average over such uncertainty. Our results demonstrate that models with a continuous P outperform those with a binary P (Fig. S5).”

Line 453 - 457

“We also assessed the performance of the models with a binary P , given by

- 1. Mode with a binary P :** *This mode calculates the Phydon prediction as:*

$$\tilde{y}_{phydon} = \begin{cases} \tilde{y}_{gRodon} & P > 0.5 \\ \tilde{y}_{phylopred} & P \leq 0.5 \end{cases} \quad 4$$

where P represents the probability that the gRodon model outperforms the phylogenetic models.”

How good does the proposed strategy generalize to the prediction of other traits?

Response: We agree with the reviewer that this approach has broad potential for application to other traits, as noted in our conclusions. While the prediction of other traits is out of scope for the current paper, work is ongoing in this area! We have added text to the introduction and conclusions to make this discussion a little more concrete with respect to existing trait prediction methods.

Line 115 – 117

“We highlight that this hybrid approach holds great potential to be extend to other trait prediction problems, such as optimal growth pH, optimal growth temperature for growth, and aerobicity.

Line 323-328

“For example, new tools have recently been developed to estimate growth temperature range, aerobicity, optimal pH, optimal salinity, and other traits from diverse genomic signals, including amino acid usage patterns [48, 53]. Each of these genomic trait predictors can potentially be embedded in a Phydon-like model that averages their output with a straightforward phylogenetic prediction to obtain a more accurate prediction.”

It seems to me that the merit of this research is predominantly making a good strategy available rather than generating algorithmic/methodological innovation. What is unclear to me how much impact Phydon has.

Response: While we agree with the reviewer that Phydon’s strategy for integrating genomic predictors with phylogenetic predictors, one which has not been used previously to our knowledge, represents a principled approach for trait prediction that likely has broad applicability across diverse trait prediction problems and will generate interest from a diverse group of microbial scientists and bioinformatics, we strongly disagree that this is the sole impact and innovation of our work.

We would like to emphasize that for many, if not most, microbial species it is simply not possible to currently measure their maximum growth rates in culture because they have not been cultured. For these organisms (the microbial majority) maximum growth rate estimates not only serve as key model parameters needed to understand community dynamics, but also give us important information about the ecological roles of these organisms. A paper’s ultimate “impact” on a field is notoriously difficult to assess, but we want to emphasize here just how useful biologically-motivated computational tools like Phydon are for probing microbial diversity across systems – their impact is seen in their use by a broad scientific community working on diverse scientific questions. For example, we received similar feedback on Phydon’s predecessor, gRodon, pre-publication. Yet this tool has been cited by hundreds of scientists working across systems as diverse as seawater, soil, and the human gut, all in a few short years. For scientific software that will be used in ways we have yet to imagine by researchers we have never met, it is incredibly difficult to assess impact.

What we can say is that a key challenge our user base currently faces is that tools like gRodon provide estimates with a high degree of variance, and thus suffer from relatively high error rates relative to, e.g., direct experimental observations. This makes it much more difficult to detect patterns in cross-species and cross-environment comparisons. To increase our power to uncover eco-evolutionary patterns across

organisms, we thus either (1) need to increase the sizes of our studies (costly and often impractical), or (2) decrease our measurement error through the development of improved predictors like Phydon (offering up to a 31% reduction in prediction error over gRodon; Fig. 2b). In a world where the cost of sequencing has dropped precipitously and large-scale sequencing studies are now the norm, we need tools that can make accurate inferences from these abundant data.

More importantly, what differentiates our work from many (though not all) trait prediction tools is our commitment to ease of use and continued maintenance. Phydon is open source, thoroughly documented, and hopefully simple enough for non-bioinformaticians to use. While this is not a feature of bioinformatics tools that is often valued in the scientific literature, it is in our view the single most important feature of a tool that people will actually use. The bioinformatics literature is littered with machine learning-based tools that no one will ever download and use. Our inclusion of ready-to-use 16S rRNA databases for trait prediction with amplicon datasets is one example of our development philosophy here. In principle, anyone could take our tool, apply it to a large database like GTDB, and then produce a 16S rRNA-to-max growth rate database to use in their own work. In practice, by providing ready-made intermediate analysis products we are able to massively expand our user base beyond a niche group of experts.

We have added text to the conclusions to further emphasize the impact of the work Lines 330-337:

“In sum, Phydon provides accurate maximum growth rate predictions for uncultured taxa, enabling scientific exploration of the ecological roles of uncultured microbial majority and their principled incorporation into ecosystem models. Phydon serves a dual role as both a powerful approach for hypothesis generation for microbial scientists working across systems (e.g., seawater, soils, and the human gut) and a key source of information for modelers who are increasingly interested in developing whole-community models of microbiome dynamics in environmental and host-associated systems [54-56].”

And have also added text to the results helping put our prediction improvements into context (Line 218-223):

“Specifically, Phydon achieves a lower MSE -- a 31% reduction in MSE -- as compared to mean square error of gRodon. We also note that the difference in performance (MSE) between Phydon and gRodon at short distances is 8.54 times the difference in performance at long distances, indicating a significant improvement at short distances while little compensation at long distances.”

There are some inconsistencies in the manuscript. For instance, I wasn't able to find the description of the analysis nor the results of the experiment referred to in Line 301,

on whether gRodon with or without temperature is better.

Response: We have had experiments with and without temperature. The gRodon estimates without considering optimal growth temperature (OGT) are generally less accurate than the ones with optimal growth temperature by assessing their MSE scores (Fig. S5). However, we implemented both temperature-free and temperature-aware prediction the package in case that users have no such data. We have added a direct comparison in the supplement (Fig. S5).

Similarly, the authors mention in Line 355 that two modes of weighing results of phylopred and and gRodon are given, however it is not described which one is used for the analysis.

Response: We thank the reviewer for their attention to detail. We have added the results that compared the geometric model and the arithmetic model. The arithmetic model is slightly better than the other in the MSE score (Fig. S5). So, **the main results are shown under the arithmetic model with a continuous P** in our study. However, the geometric mean is more reasonable for the exponential doubling time/growth rates. Thus, we provided the two modes options in our package. We also examined the performance of the arithmetic model with binary P. The results show instability at some cutting time points where the CUB and Phylopred model performed similarly but bias in different direction. We added a discussion on that.

Line 429 – 435

“We suspected that the geometric mean model would be more suitable for averaging growth predictions given its usual application for analyzing percentage changes and positively skewed data where it provides greater accuracy. Nevertheless, our results show that both weighted prediction modes performed similarly with the arithmetic model is slightly better in its MSE score (Fig. S5). So, we presented the main results using the arithmetic mean model and set this as the default mode in the Phydor R package.”

Minor comments

line 123: maybe the authors could describe the dataset in a few words, just using "our dataset" is a bit vague and doesn't help much

Response: We have now incorporated a brief description of our dataset at the top of the result, with a more detailed description available in the methods section.

Line 121 – 124:

“We compiled a dataset of 633 species with recorded doubling times from the Madin et al. trait database [9]. However, 85 species were excluded due to unidentifiable species names in the Genome Taxonomy Database (GTDB). As a result, our final

dataset comprised 548 species (Fig. S2)."

line 187: Yes, Phydon seems to have lower variance than Phylopred/NNM, but only barely so for gRodon, maybe some numbers would be helpful to drive home this point (as the difference in total between Phydon/gRodon is difficult to see on the diagram)

Response: Indeed, we agree with the reviewer that Phydon has only slightly different variance from gRodon, but a larger decrease in variance relative to other phylogenetically-aware prediction tools. In this sense, Phydon gains the advantages of a phylogenetic prediction approach without the cost of increased prediction variance. We have revised the text to reflect this point:

Line 223 – 226

"Additionally, the variance of the Phydon's predictions is lower than that of alternative phylogenetically-aware prediction approaches, although the variance of the gRodon model is nearly indistinguishable from that of the Phydon model (Fig. S1)."

line 280-282: swap numbers 548 and 633 -> 548 - 85 != 633

Response: We thank the reviewer for their attention to detail. Madin et al.'s data has 633 species. We discarded 85 species as they are unidentifiable in GTDB. Thus, we have 548 species as our data set. We have clarified it in the text.

Line 121 – 124:

"We compiled a dataset of 633 species with recorded doubling times from the Madin et al. trait database [9]. However, 85 species were excluded due to unidentifiable species names in the Genome Taxonomy Database (GTDB). As a result, our final dataset comprised 548 species (Fig. S2)."

line 283: should probably be Fig. S2, not S1

Response: We thank the reviewer for their attention to detail. We have corrected this error.

Fig. 3/4c): sometimes phylogenetic distance axis goes from high to low, sometimes the reverse -> could it always go from low to high?

Response: We thank the reviewer for this helpful suggestion. In Fig. 3, the x axis represents the cutting time from past to present, which is consistent with phylogenetic time. In Fig. 4c, the x axis is the phylogenetic distance between the focal species and others. Thus, they are different. We have added a figure (Fig. 2b) to clarify this difference. We also unified the x axes that indicate the time from past to present for Figures 2 and 3.

Questions

Fig. 2a) Why doesn't this go to 0.14, as all the other figures? How do they get values for such low distances? How many clades are there in this case?

Response: We thank the reviewer for pointing out an item that would likely have confused our readers. The distance on the x axis of Fig 3a is the cutting time point D_c at which the phylogenetic tree is divided to n clades while the other is the average phylogenetic distance D_p between the focal species and the 5 nearest species. Given a cutting time point D_c , the average phylogenetic distance D_p between each query species and the nearest 5 species in the training data can vary. Generally, we show in Fig. 2a that the minimum average phylogenetic distance between test and training data decreases when the tree is separated into more clades. We have added one figure (Fig 2b) to clarify those metrics and changed the axis labels accordingly.

line 105-119: paragraph also contains results? Is it supposed to be like this?

Response: We have revised this paragraph to remove specific references to results as requested.

line 195: I don't get this reasoning: the threshold estimate is unclear, because gRodon/Phylopred predictions are close to each other (and also because of the data) -> so a binary switch would be just better?

Response: Please see our detailed discussion of the threshold-based model, which has been added to the package, above.

line 242/Fig. 4b): If this diagram shows that Phydon is better than gRodon, doesn't it also show that Phylopred is better than Phydon?

Response: This figure panel does not speak to differences in prediction quality across the three methods, but simply shows the distribution of those predictions across species in GTDB. We do appreciate that our statement about growth classes, which was meant more to discuss overall differences in the shapes of these distributions, could also be taken to have some implicit value for ranking these methods. We have revised this sentence to avoid any such confusion:

Line 285-286

"Interestingly, Phydon and Phylopred both seem to distinguish these major growth classes much more readily than gRodon (Fig 4b)."

line 243-246/Fig. 4c): Why this sentence/figure? Isn't it by design intended that Phydon uses gRodon for distantly related organisms? Wasn't this designed this way? Is it just regression?

Response: We agree that this pattern is expected as it is “baked-into” the model and thank the reviewer for pointing out a sentence that could have caused some confusion. We have revised the sentence for clarity:

Line 286-289

“As we would expect, the predictions of Phydon and gRodon converge as phylogenetic distances increase (Fig. 4c), showing how Phydon increasingly relies on genomic factors, particularly codon usage bias (CUB), for extrapolation.”

line 269-277: This paragraph seems off, particularly the first two sentences do not seem to fit.

Response: We are not entirely sure we understand the reviewer’s specific critique here, but we have revised this paragraph slightly for flow. Hopefully the relevance of the point is now clear.

Line 303 – 307

“However, our method is not without limitations. A key challenge lies in its performance when applied to taxa that have undergone rapid trait evolution. Periods of accelerated trait evolution, such as those occurring in rapidly changing environments, can weaken the predictive power of both phylogenetic models and models based on genome-scale evolutionary patterns like CUB. These models rely on the assumption of relatively stable evolutionary processes [52], and this represents an ultimate limitation of all genome-based trait prediction models.”

line 335: Does phylogenetically blocked cross-validation do multiple tests on the same n ? So unlike normal cross-validation?

Response: Phylogenetically blocked cross-validation works the same as n -fold cross validation, except that the assignment of data to folds is done according to phylogenetic groupings rather than a random draw. Some authors additionally iterate when using n -fold cross validation (repeatedly drawing new fold assignments), though that is not the procedure we use here. While we are not sure, we think the reviewer is additionally asking about how our cross validation procedure works when changing the point at which we cut the tree (D_c in Fig 2). Here, we run phylogenetically blocked CV at each value of D_c , so that the number of folds, n , will change as well as the assignment of observations to folds. Thus, the same organisms will be grouped differently into folds depending on the height at which the tree is cut. We have added text to explain these points in the main text:

Line 140 - 147

“A cutting time point D_c , at which the tree is divided into several clades, is identified based on the desired number of clades, n (Fig. 2). Cutting the tree closer to the present results in a greater number of clades with smaller phylogenetic distances

between them, while cutting further in the past at larger time points produces fewer clades with greater phylogenetic distances (Fig. 2a). This cutting time point thus serves as a proxy for the phylogenetic distance between training and test clades. This is a form of phylogenetically blocked cross-validation, wherein observations are grouped into folds following their evolutionary relationships rather than at random [39].”

Due to these issues, we would suggest to reject the paper. Particularly, the small benefit of Phydon, combined with the small amount of innovation motivates our recommendation.

Reviewer #3 (Remarks to the Author):

Response: We thank the reviewer for their time and contributions to our manuscript revision.

Reviewer #4 (Remarks to the Author):

Xu et al. propose a new method to predict microbial growth rates using both codon usage and phylogenetic information.

This approach improves upon a previous method gRodon (also developed by the last author), and is applied to a large number of genomes to generate a free to use database.

The method is very well documented online. I have not tried it, but the documentation seems very detailed.

Response: We thank the reviewer for their summary and detailed comments.

The manuscript is clear and easy to follow. I have one major criticism to make, and provide some comments below that I think could help improve the manuscript.

My major criticism is methodological. First, the authors have performed cross-validation to evaluate the performance of the two base methods. It was unclear to me whether, for a given n (the number of clades), several predictions were made, where a different clade is hold out each time.

Response: Phylogenetically blocked cross-validation works the same as n -fold cross validation, except that the assignment of data to folds is done according to phylogenetic groupings rather than a random draw. We additionally change the point at which we cut the tree (D_c in Fig 2), running a cross-validation procedure at each cutting height. So, we run phylogenetically blocked CV at each value of D_c , so that the

number of folds, n , will change as well as the assignment of observations to folds. Given a specific value of n (corresponding to a specific cutting height D_c), our procedure then works the same as typical blocked-cross validation where each of the n folds is held out in turn as a test set and model performance is averaged over the n models built as a result of the procedure. In other words, we do exactly as the reviewer suggests. For example, if $n=10$, we performed 10 times of our approach to train 10 models and applied to 10 corresponding testing data sets. The final mse score is calculated using all results at this cutting time point. We have further clarified this in the text.

We have added the explanation and Fig. 2b for clarification.

Line 140 - 147

“A cutting time point D_c , at which the tree is divided into several clades, is identified based on the desired number of clades, n (Fig. 2). Cutting the tree closer to the present results in a greater number of clades with smaller phylogenetic distances between them, while cutting further in the past at larger time points produces fewer clades with greater phylogenetic distances (Fig. 2a). This cutting time point thus serves as a proxy for the phylogenetic distance between training and test clades. This is a form of phylogenetically blocked cross-validation, wherein observations are grouped into folds following their evolutionary relationships rather than at random [39].”

Second, no cross-validation was used to evaluate the performance of the combined approach. I believe common practice would be to leave out some data to train the two base methods, that could then be used to evaluate the performance of the combined approach. Otherwise, there is a risk of overfitting.

Response: We agree with the reviewer that this would be a better approach to model fitting and have modified our analysis accordingly. We now report the cross-validated performance of Phydon in all manuscript figures (incl. Fig 2). For a given n , we completely exclude a target species and train models from the rest data. We further compared predictions from three modes, i.e., the geometric model, the arithmetic model with binary P and the arithmetic model with continuous P . We found that they performed similarly except the arithmetic model with binary P shows instability at some cutting time points. Please see the new result (Fig. S5).

Minor comments:

p6 : "provides accurate directional prediction" : what does directional mean here?

Response: We agree this wording was confusing. We meant to say that the gRodon model can generally distinguish between fast and slow-growing species. We have clarified that in the text.

Line 158 - 160

“The gRodon model generally distinguishes fast and slow-growing species. Its performance is consistent across the tree of life, as demonstrated by a stable mean squared error across varying phylogenetic distances (Fig. 3a).”

I169: "we observed divergent performance patterns between the gRodon and Phylopred models for fast-growing and slow-growing species (Fig. 3bc)": this is a surprising pattern, I would have expected the opposite: that the signal of selection would be strong for species with high growth rate, which gRodon could pick. For slow-growing species, I would expect little signal in codon usage, and therefore more errors for gRodon. Could the authors provide some insight into what may be happening?

Response: We thank the reviewer for their close reading of our work. We are also surprised that both methods have severe limitations when applied to slow-growing species. We don't have a clear answer here – there are couple explanations we can think of: (1) that slow growth evolves sporadically (and quickly) across the tree of life making phylogenetic prediction untenable (see our point in the conclusions about evolutionary rates), (2) that the growth rates of slow growing species have not been measured accurately due to the difficulty of cultivating these organisms (garbage in, garbage out). These issues are made more severe by the lack of cultured slow-growing taxa in general available to train a model on (one of the key findings of the original gRodon paper).

Line 180 – 202

“Interestingly, we observed divergent performance patterns between the gRodon and Phylopred models for fast-growing and slow-growing species (Fig. 3bc). For slow-growing species, the gRodon model consistently outperforms the Phylopred model across all phylogenetic distances (Fig. 3c). In contrast, the Phylopred model shows superior performance over the gRodon model for fast-growing species as the phylogenetic distance decreases (Fig. 3b). These findings suggest that, for fast-growing species, phylogenetic relationships capture selective signals more effectively than codon usage bias, which is inherently limited as a genomic statistic for explaining maximum growth rates [41]. However, both methods face challenges in predicting traits for slow-growing species. One challenge is the difficulty of culturing these species and accurately measuring their maximum growth rates, leading to insufficient and potentially low-quality data for model training.

The lower mean squared error (MSE) achieved by the gRodon model compared to the Phylopred model may be due to the distribution of slow-growing species across the phylogenetic tree. These species exhibit weak phylogenetic signal, which challenges Phylopred's reliance on evolutionary relationships. In contrast, the CUB patterns from slow-growing species in the training dataset remain informative for predicting traits in the test dataset, enhancing gRodon's performance.”

l196: "we disfavored such an approach.": it would seem like this paragraph does not quite correspond to what the authors used, i.e., the geometric model with a logistic regression to infer the P parameter. I think some rewriting would clarify what the authors really did. In addition, a plot of the P value on the y axis, and the phylogenetic distance on the x axis, could be useful to understand the behaviour of the model.

Response: We have rewritten this paragraph for clarity. Additionally, we have added a plot as suggested to show how the relative expected accuracies of gRodon prediction and phylogenetic prediction change with both the phylogenetic distance of the query genome to the training set and the expected growth rate of the query genome (Fig. S6).

Line 228-239

"We defined the weight parameter P as a continuous value between 0 and 1 (see Method), ensuring that Phydon estimates always incorporate information from both phylogenetic relationships and genomic statistics. The parameter P can also be treated as a binary variable, selecting the method that achieves higher accuracy at a given phylogenetic distance. However, similar to known statistical challenges with piecewise regression this approach introduces instability in overall performance due to uncertainty when estimating the appropriate threshold value for switching between models (Fig. S5). Continuous weighting schemes (as used above) average over such uncertainty. Our results demonstrate that arithmetic models with a continuous P outperform those with a binary P (Fig. S5). Alternatively, the binary P approach leads the model to discard information from one source entirely, favoring either phylogenetic relationships or genomic statistics, but never both. Thus, we disfavored such an approach."

l280: "We compiled a training dataset of 548 species with recorded doubling times from the Madin et al. trait database, after discarding 85 species due to unidentifiable species names in the Genome Taxonomy Database (GTDB), resulting in a total of 633 species [9] (Fig. S1)." : this sentence is not clear and makes it look like you ended up with 633 species but started with 548 species. Also, I believe you want to refer to Fig S2, not S1.

Response: We thank the reviewer for their attention to detail. We have fixed these errors and clarified the size of the dataset.

Line 121 – 124

"We compiled a dataset of 633 species with recorded doubling times from the Madin et al. trait database [9]. However, 85 species were excluded due to unidentifiable species names in the Genome Taxonomy Database (GTDB). As a result, our final dataset comprised 548 species (Fig. S2)."

l306: "To balance information content and computational efficiency, we performed

random sampling, selecting up to five genomes per species in our training set ": If I understand correctly, this means that the authors are assuming that all strains of a species, despite having some differences in their genomes, share the same growth rates. Are there reasons to expect that this is indeed the case, or should this be mentioned as a caveat? Some discussion would be useful.

Response: Indeed, this is a necessity since our growth rate data comes at the species level and this is also the level at which gRodon is trained. We will note that growth rates tend to be strongly conserved up to the genus level, meaning that intra-species variability is a relatively small contributor to these patterns.

Line 373 – 376

"We assume that highly similar genomes share traits. This assumption is essential to our methods and also to other prediction methods that use genomic statistics, where it has been shown that maximum growth rate is strongly conserved up to approximately the genus level [11, 12]."

l354: "Phydon operates in two modes:" : I have not seen a comparison of the two approaches; perhaps a figure in the supplementary material would be useful.

Response: We agree that such a comparison is necessary and we have added the comparison (Fig. S5).

Fig 4b: Phlopred -> Phylopred

Response: We thank the reviewer for their attention to detail. This has now been fixed.

REVIEWERS' COMMENTS

Reviewer #1 (Remarks to the Author):

We thank the authors for addressing all of our comments and for further explaining how the phylogenetic tree was divided and how the parameter P affects the behavior of the model. Overall, the authors have improved the quality of the manuscript. Therefore, we endorse the current version for publication, provided the authors address two additional requests.

We thank the reviewer for their endorsement and detailed comments throughout the review process, which we believe have greatly improved the manuscript.

1. Adding Figure S6 to the GitHub repo will help users identify the optimal value of P for their analyses.

We have added the figure to the Github repo and README.

2. Can the authors state in the main text the purpose of the inset from Figure 3A? The MSE of Phydon is lower than that of gRodon at $D_c = 0.07$. Therefore, it is unclear to me why the authors added the inset. What message are they trying to convey?

Thank you for the feedback. The addition of the inset figure was made in response to Reviewer 2's suggestions during the previous round of revision, aimed at highlighting Phydon's improvement over gRodon at the smallest phylogenetic distance. We have clarified this more explicitly in the revised manuscript.

Line 181 – 183:

"At the smallest cutting time ($D_c = 0.07\text{my}$), Phydon reduced the median squared error for species with doubling times under 30 minutes by 22.4% compared to gRodon (Fig. 3a, inset)."

Reviewer #1 (Remarks on code availability):

Sound.

Reviewer #2 (Remarks to the Author):

The authors substantially revised the manuscript and addressed our concerns adequately. We have only one further minor remark.

We thank the reviewer for their endorsement and detailed comments throughout the review process, which we believe have greatly improved the manuscript.

Minor comments

General remarks on the applicability of the approach to further traits let us wonder, what evidence the authors provide towards this claim. Now, a new sentence introduced in the manuscript in lines 115-117 reiterates this possibility, but unless the authors produced (preliminary) experimental evidence, it would be best to limit references to future work to Conclusion Section. While the translation to other traits is fitting for an outlook, mentioning this in the introduction could be misleading, the presented work does not as it is not the main part of the work.

We thank the reviewer for their attention to detail. We agree that this is more appropriately discussed exclusively in the conclusions (where it is also brought up) and have removed this sentence from earlier in the manuscript.

Reviewer #3 (Remarks to the Author):

We thank the reviewer for their efforts on this manuscript, which we believe have greatly improved the work.

Reviewer #3 (Remarks on code availability):

The code is well-documented and provides examples of usage.

Reviewer #4 (Remarks to the Author):

Xu et al. have satisfactorily addressed my comments, and I have no further major comments, and only minor elements below.

I congratulate the authors on their work.

We thank the reviewer for their endorsement and detailed comments throughout the review process, which we believe have greatly improved the manuscript.

Minor comments:

l75: "where traits values can be estimated": trait values

l116: "optimal growth temperature for growth": too many "growth"

l222: "at short distances while little": with little

l423: "with the arithmetic model is slightly better in its MSE score (Fig. S5)": remove "is"

We have revised the manuscript accordingly.